# Low Temperature Decomposition of Polystyrene

**Hideki Kimukai** [1], **Yoichi Kodera** [2], **Koushirou Koizumi** [3], **Masaki Okada** [4],
**Kazunori Yamada** [4], **Toshihiko Hiaki** [4] and **Katsuhiko Saido** [1,4,*]

[1] Albatross Alliance, 2234-1 Minamiboso, Shirahama, Chiba 2950102, Japan; hkimukai@hotmail.com
[2] National Institute of Advanced Industrial Science and Technology (AIST), Tsukuba, Ibaraki 3058569, Japan; y-kodera@aist.go.jp
[3] College of Science & Technology, Nihon University, Funabashi, Chiba 2748501, Japan; Koizumi.koushirou@nihon-u.ac.jp
[4] College of Industrial Technology, Nihon University, Narashino, Chiba 2758575, Japan; okada.masaki@nihon-u.ac.jp (M.O.); yamada.kazunori@nihon-u.ac.jp (K.Y.); hiaki.toshihiko@nihon-u.ac.jp (T.H.)
[*] Correspondence: katsu.saido@gmail.com

**Abstract:** Styrene oligomers (SOs), of styrene (styrene monomer, SM), 1,3-diphenylpropane (styrene dimer, $SD_1$), 2,4-diphenyl-1-butene (styrene dimer, $SD_2$) and 2,4,6-triphenyl-1-hexene (styrene trimer, ST), had been detected in the natural environments far from industrial area. To confirm SOs formation through thermal decomposition of polystyrene (PS) wastes in the nature, purified polystyrene (SO-free PS) has been shown to decompose at 30 to 150 °C. The SO ratio of SM:SD:ST was about 1:1:5 with ST as the main product. Mass spectrometry with selected ion monitoring was used for the quantitative analysis of the trace amounts of SOs. The rate of PS decomposition was obtained as $k\left(\text{year}^{-1}\right) = 5.177\ exp(-5029/T(\text{K}))$ based on the amount of ST. Decomposition kinetics indicated that not only does drifting lump PS break up into micro/nano pieces in the ocean, but that it also subsequently undergoes degradation into basic structure units SO. According to the simulation at 30 °C, the amounts of SOs in the ocean will be over 400 MT in 2050.

**Keywords:** low-temperature decomposition; polystyrene; styrene oligomer; plastic debris; chemical contamination

---

## 1. Introduction

Total plastic production in the 1950s has been shown to be around several million metric tons (million MT), but at present, cumulative production has been estimated as $6.98 \times 10^9$-MT during the period 1950 to 2015 [1,2]. Waste plastic from land sources is continually flowing into world oceans via rivers due to accidents or carelessness. In 1972, Carpenter and Smith pointed out marine contamination from plastics drifting on the surfaces of the Sargasso Sea [3]. In 2001, Moore et al. [4] reported debris plastics to form garbage patches in the Pacific Ocean and to increase by 17-fold (by weight) or 95 times (by pieces) as much the amounts of contaminants in the past 30 years. Thompson et al. [5], Takada [6], Lavender [7] and Isobe et al. [8] have shown this drifting plastic to break up into a great many small pieces by the action of waves and effects of light to form micro/nanoplastics which in turn lead to massive plastic contamination in bulk. All drifting plastic quite likely break up into small pieces in this manner [9]. By 2050, debris plastics will have attained a weight exceeding that of all fish throughout the oceans of the world [10].

The authors have been engaged in the collection of coastal waters and beach sands from around the world for the past 20 years so as to determine what chemicals are generated from plastics in oceans worldwide [11–15]. However, the manner in which various chemicals are derived from plastics has

yet to be fully clarified. Plastics decomposition has long been a topic of intensive research [16–22] at temperatures of 250 °C or higher.

In order to estimate the amounts of SOs from drifting PS in the ocean, the rate of PS decomposition needed to be determined at a living temperature range in the nature. However, to date, no kinetic research on PS decomposition has been conducted at the lower temperature range in the natural environment. Plastic debris undergoes degradation called as weathering in the ocean and on a beach, being exposed to salt water or to the sun light in the presence of the air. Temperature is an essential factor among various potential factors to govern plastic degradation. PS decomposition at low temperature was thus examined in the present study using polyethylene glycol (PEG1540) as a heating medium.

In this study, kinetic parameters of PS decomposition at a lower temperature range, 30–150 °C, was determined to obtain the rate of SOs formation from PS. The amounts of SOs and PS in the ocean with time were simulated based on the kinetic parameters obtained although there are various factors other than reaction temperature.

Using purified PS without contamination of SOs, low-temperature decomposition was carried at 30 to 150 °C. The products obtained were styrene oligomer (SOs), of styrene (styrene monomer, SM), 1,3-diphenylpropane (styrene dimer, $SD_1$), 2,4-diphenyl-1-butene (styrene dimer, $SD_2$) and 2,4,6-triphenyl-1-hexene (styrene trimer, ST). The composition ratio of SM:SD:ST was 1:1:5 and the main product was ST. The rate of ST formation by PS decomposition was measured at 30 to 150 °C. Kinetic parameters of the PS decomposition were determined and the activation energy of the conversion of PS into ST was given as 45.0 kJ mol$^{-1}$. One MT of PS was found to decompose at a rate of 0.3 g per year at 30 °C.

Simulation results indicated that the total amount of degraded PS in the ocean has been as much as 430-MT during the period, 1950 to 2050. The results indicated PS to have little stability toward heat, kinetically. Drifting macro PS not only breaks up into micro/nanosized pieces, but subsequently degrades into basic structure units of SOs in the ocean.

## 2. Experimental

### 2.1. Preparation of Reaction Samples

Commercial pellets of PS (Number–average molecular weight: 500,000, Teijin Chemicals, Ltd., Tokyo, Japan) were found to contain 90 mg kg$^{-1}$ of unreacted SOs along with various additives. For the removal of these chemicals, 50 g PS were dissolved in 2000 mL benzene and reprecipitated with 4000 mL methanol at room temperature. This operation was conducted three times and SO remained at less than 0.1 mg kg$^{-1}$. Purified PS was used in this operation. Figure 1a shows the procedure for purifying PS and Figure 1b shows the subsequent treatment.

### 2.2. Reagent

Polyethylene glycol (PEG1540, Average molecular weight 1350–1650, Wako Pure Chemical Co., Osaka, Japan) was used as the heat medium due to high solubility for PS, thermal stability and low volatility. Benzene and methanol for dissolving PS or reprecipitation and tetrahydrofuran (THF) as the GPC eluent all these were of reagent grades and manufactured by Wako Pure Chemical Co. $SD_2$ and ST were prepared by the decomposition of PS and purified by distillation under reduced pressure by boiling point fractionation [23]. The purity of $SD_2$, ST was determined by gas chromatography (GC) with a flame ionization detector (FID) as 99.8% or more before use. The internal standard, phenanthrene, and diphenyl were used after being purified using a special grade of Kanto Chemical Co., Inc. (Tokyo, Japan) following sublimation treatment. SM and $SD_1$ (Wako Pure Chemical Co.Osaka, Japan) were used after the distillation of reagents prior to use.

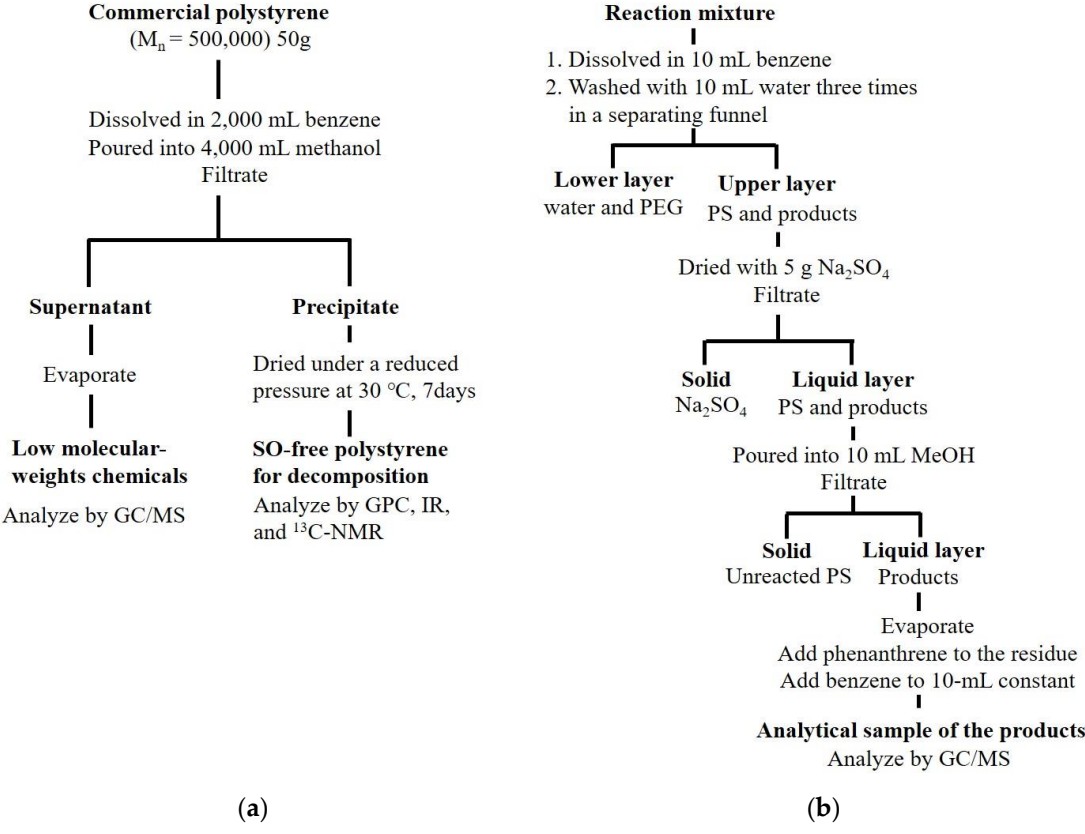

**Figure 1.** Procedure for purifying (**a**) styrene oligomers (SO)-free polystyrene (PS) and (**b**) subsequent treatment.

*2.3. Decomposition Method*

Figure 2 shows the schematic diagram of the experimental equipment for PS decomposition higher than 50 °C. Silicone oil bath was used for heating. A 20-mL glass flask was used for SO-free PS decomposition. Chromyl/alumel thermocouples were used to monitor the temperature of a reaction mixture in the flask and that of an oil bath. The flask has a nitrogen gas inlet and a gas seal for nitrogen gas flowing out. A heat medium, 4.9 g of PEG a stirrer were placed in the flask and nitrogen gas was introduced at a rate of 50 mL min$^{-1}$.-When the flask had reached a predetermined temperature, 0.1 g SO-free PS (cut off small pieces under 5 mm) and diphenyl as a surrogate ($1.0 \times 10^{-6}$ g in benzene) were charged into the flask. The reaction solution was stirred at 500 rpm. Temperature was adjusted to a predetermined temperature ± 1 °C using a digital thermometer.

PS decomposition at 30 °C was carried out without solvent because the melting point of PEG1540 is about 45 °C. PS samples were stored in a flask placed in a thermostatic chamber (Yamato Scientific Co., Ltd.,Tokyo, Japan) control accuracy ±1 °C) under nitrogen gas. The reaction at 30 °C was continued for three years.

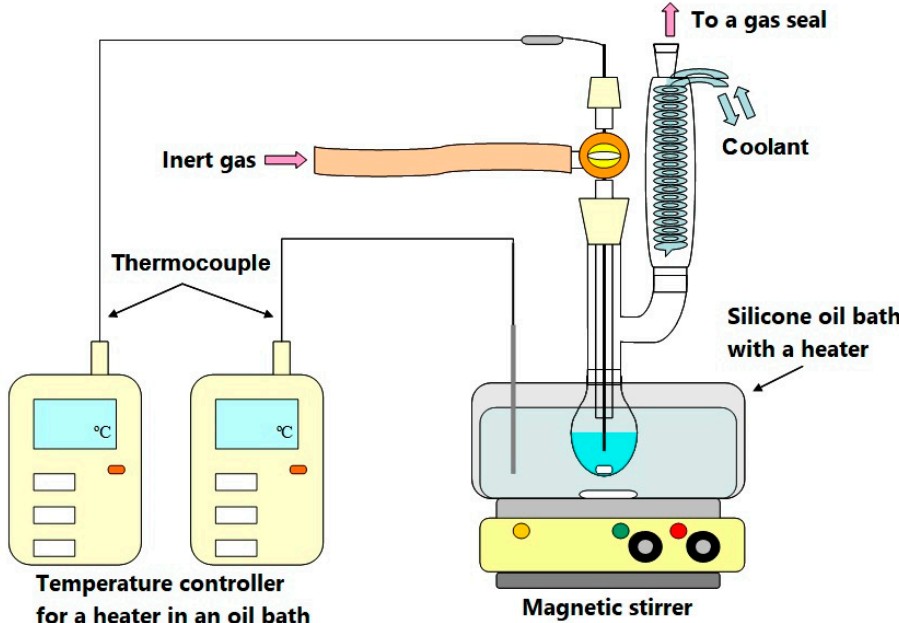

**Figure 2.** Schematic diagram of a decomposition equipment.

### 2.4. Apparatus and Operating Conditions

SO-free PS was heated at a fixed temperature for a given period in a flask. The reaction mixture was recovered with 10 mL benzene and was transferred to a 50 mL separating funnel and washed three times with 10 mL purified water to remove PEG. After drying with 5 g anhydrous sodium sulfate overnight, the solution was filtered and transferred to 10 mL methanol to precipitate unreacted PS. The solvent was completely removed from the solution by evaporation at 25 °C and the residue was collected with benzene in a 10-mL volumetric flask. Following the addition of phenanthrene, 1 μL of the solution was injected into GC/MS with a micro-syringe and analyzed. The details are shown in Figure 1b. The GC used was HP6890 (Agilent Technologies, Inc., Santa Clara, CA, USA). The mass spectrometer (MS) was JMS-AMII manufactured by JEOL, Ltd. (Tokyo, Japan) and the separation column was DB-1 manufactured by Agilent J & W. The operating conditions are shown in Table 1.

**Table 1.** Apparatus and operation conditions of GC/MS analysis.

| | |
|---|---|
| Apparatus | JEOL JMS-AM II with GC/MS (HP6890) |
| Selected Ion (SI, *m/z*) | 78, 104, 105, 152, 178, 193, 196, 207, 208, 312 |
| Injection | 1 μL |
| Column | DB-1, L 30 m, ID 0.32 mm, Thick 0.25 μm |
| Injection method | Spitless |
| Injection temperature | 250 °C |
| Column temperature | Initial temp. 40 °C (holding 5 min), max temp. 290 °C (holding 5 min), program rate 15 °C/min |
| Interface temperature | 250 °C |
| Ion source temperature | 200 °C |
| Ion acceleration current | 70 eV |
| Current | 300 μA |
| PM voltage | 600 V |
| Carrier gas | He, 1.4 mL/min |

## 3. Results and Discussion

### 3.1. Accuracy

The mass spectrum obtained by GC/MS, TIM was used to determine fragment ion (*m/z*) of the target chemicals (SM, $SD_{1,2}$, ST). The quantitative analysis was conducted by using the monitoring ion: Q (quantitative ion), *m/z*: 104, 154, 178, 196, 208, 312 and qualitative ion: *m/z*: 78, 105, 117, 152, 193 ions. Calibration curves were prepared by the internal standard method using the detected ion peak area ratio (I/Q). For quantitative analysis of the target chemicals by mass spectrometry, Selected Ion Monitoring (SIM) was used because target detection performance of SIM was 100 times higher than that in Total Ion Monitoring (TIM). It was thus possible to obtain very low concentrations ($10^{-9}$ level) of SOs by the SIM.

A standard solution of each compound (SM, $SD_{1,2}$, ST) was prepared to obtain a target solution using a pipette and volumetric flask, and the calibration curve of each compound was prepared. Curve linearity ranged from 0.2 µg to 10 mg $kg^{-1}$ with a correlation coefficient, r = 0.9996 to 0.9999. The detection limit was found at S/N = 2 to be 10 µg $kg^{-1}$.

### 3.2. Effects of Temperature and Time

Polymer decomposition such as that for PS has long been studied [16–22]. Since the polymer has low thermal conductivity, a preliminary heating time is necessary to achieve the target temperature for decomposition [21]. The generated SOs has an extremely low value of less than µg $kg^{-1}$ thus making analysis difficult in this study, this difficulty was overcome by a new decomposition using PEG as the heat medium. Among the various factors governing plastic degradation in the nature, we focused thermal effect on PS decomposition in the experiments. We assume first-order reaction of PS to form SOs. The rate of ST formation was calculated at each reaction temperature. ln *k* and the reciprocal of the absolute temperature ($T^{-1}$) are shown in Figure 3, which shows a linear relationship in the entire temperature range of 30 to 150 °C. PS activation energy of 45 kJ $mol^{-1}$ was obtained from the slope of the line by using the Arrhenius equation. A low activation energy 45.0 kJ $mol^{-1}$ indicated a secondary reaction, a backbiting reaction [24] to be a dominant factor in low temperature decomposition.

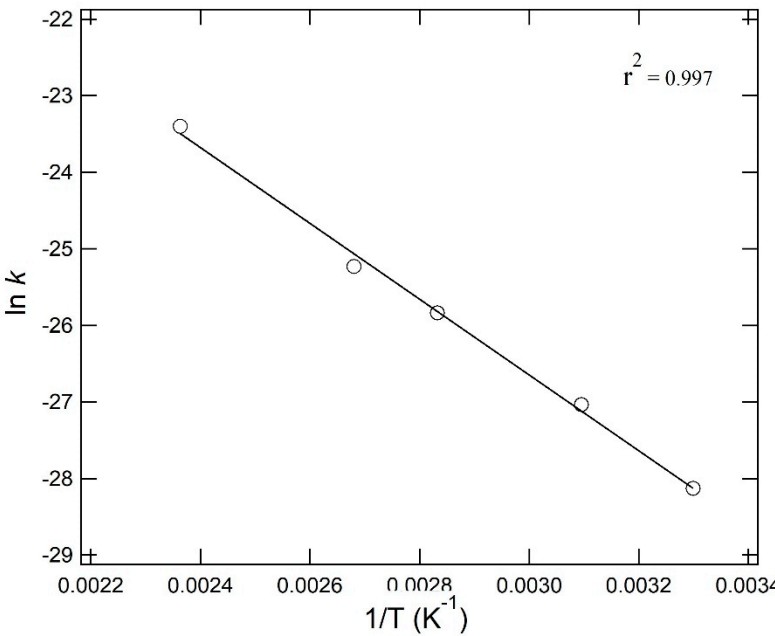

**Figure 3.** Arrhenius plot of styrene trimer (ST) formation through PS-decomposition at 30 to 150 °C.

SO composition did not change in the range 30 to 150 °C. The largest component was ST followed by SD and SM. The SO composition ratio of SM:SD:ST was about 1:1:5. Trace amounts of ethylbenzene, propyl benzene and benzaldehyde were observed along with secondary reactions when temperature reached 200 °C and higher. After the reaction, the PS was recovered and prepared into a film and analyzed by IR (infrared absorption spectrum). The IR analysis showed no change in the PS main chain (2924, 2850 cm$^{-1}$ aliphatic stretching vibration, 1450 cm$^{-1}$ aliphatic vibration). Alternatively Gel Permeation Chromatography (GPC, Column: Asahipak GF-7MHQ, Detector: RI, eluent: THF 0.6 mL/min, Sample: 2 mg in 100 mL THF, 20 μL injection, 30 °C) detected number–average molecular weight decreased by 20% when reaction conditions were between 200–280 °C for 30 min or more [21].

### 3.3. Degree of PS Degradation in Ocean, Determined from Rate Constant of ST Formation

Various chemical species possibly derived from artificial sources have been detected in the nature. Figures 4 and 5 show typical TIM-GC/MS chromatograms of analytical samples by extraction of sand and pebble in each sampling site far from industrial area.

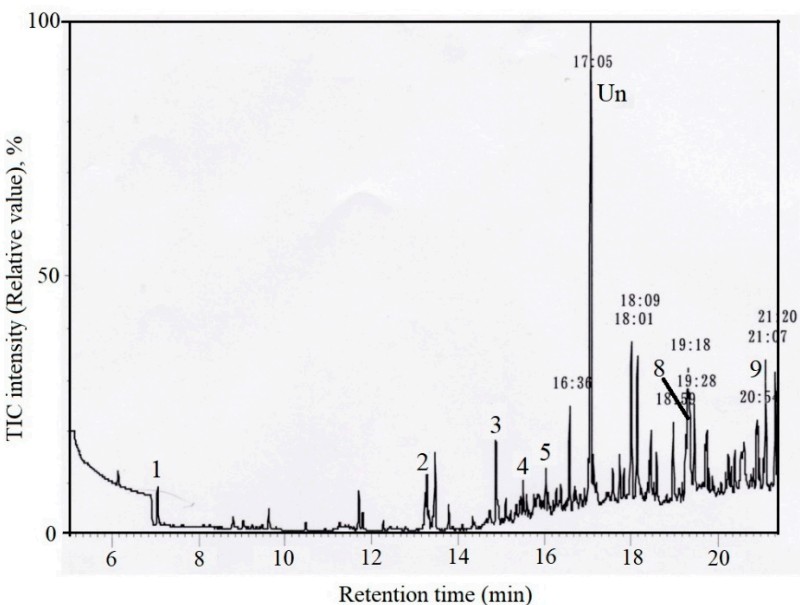

**Figure 4.** TIM-GC/MS chromatograms of the sample extracted from sand in Waikiki beach, Hawaii, 21°16' N, 158°8' E, Date: 25 March 1975.

Figure 4 shows the chromatogram of an extraction sample of sand from Waikiki beach at Oahu, Hawaii in 1975 and Figure 5 shows the chromatogram of an extraction sample of pebbles by Showa base, Antarctic Continent in 1984. Compared with the mass fragments of each standard sample, Peak 1 was identified as SM, 3 as diethyl phthalate, 4 as SD$_1$, 5 as SD$_2$, 8 as bisphenol A, 9 as ST. 2 as diphenyl added as a surrogate and 6 as phenanthrene added as an internal standard and Un means unknown chemicals. SM has been shown a breakdown product formed by cinnamon mold flora and possibly may be present in oceans as a single contaminant [25]. However, the other styrene oligomers are not naturally present in ocean. Considering the constant ratio of SM:SD:ST, the authors concluded that SOs have been shown to be the degradation products of PS from land-based sources. The chemical contamination generated from PS had already been present in the nature over 45 years ago.

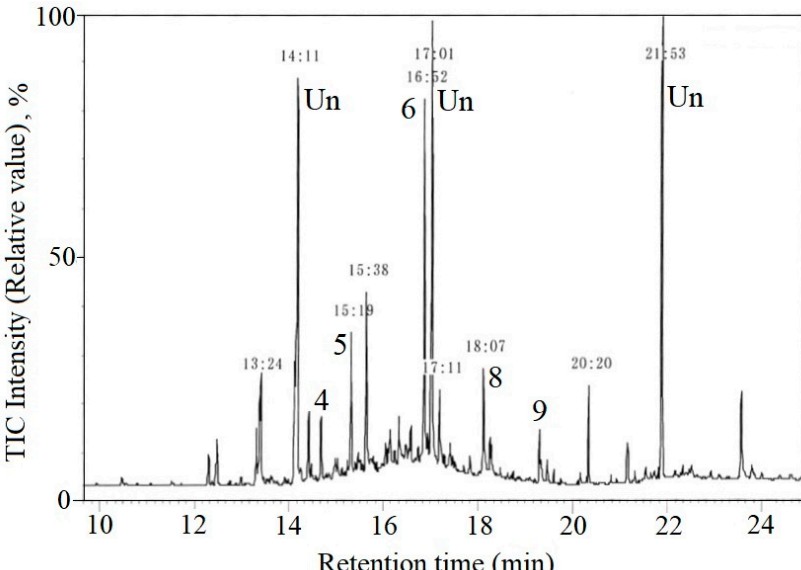

**Figure 5.** TIM-GC/MS chromatograms of the sample extracted from pebble by Showa base, Antarctic continent, 60°00′ S, 39°35′ E, 1984.

Jambeck et al. [26] estimated total waste plastics inflow in 2010 into world oceans to range from 4.8 to $12.7 \times 10^6$ MT. However, in all these computations [3–6,26], there has always been considerable margin for error. There is really virtually no completely relivable information on the quantities of plastics that have undergone degradation and sedimented to the ocean floor.

The rate of ST formation was thought same as the rate of PS decomposition at ambient temperatures in oceans and on beaches since PS decomposition at low temperature, 30–150 °C, give ST as the dominant product [22]. The decomposition rate, $k$, of PS (basically the same as the formation rate of ST) was found to be $S_t = P_0[1 - exp(-kt)] \approx P_0kt$, where $k(min^{-1}) = 9.850 \times 10^{-6} exp(-5029/T)$ where $P_0$ is the initial amount of purified PS in a flask, $t$ the reaction time and $T$ the temperature in Kelvin. Rate of ST formation was $6.15 \times 10^{-13}$ min$^{-1}$. Converting the unit of rate into annual rate, the annual decomposition rate, $k$, of PS was given as $k(year^{-1}) = 5.177 exp(-5029/T)$ where $P_0$ is the amount of PS in the ocean, $t$ time in years since 1950 and $T$ the temperature in Kelvin. The ST rate was $3.233 \times 10^{-7}$ year$^{-1}$ at 30 °C. One MT PS decomposed at a rate of 0.3 g per year at 30 °C. PS production of seven percent in the total plastics production [2]. Jambeck et al. assumed three percent of the total plastic consumption as plastic inflow to the ocean [26]. The cumulative amounts of PS and SOs in the ocean were simulated as shown in Figure 6.

The simulation was conducted by using three differential equations (Equations (1) through (3)). The production share of PS at seven percent and inflow ratio at three percent were assumed to be constant during 1950 to 2050.

$$w(t) = 1.50 \times 10^6 + 4.845 \times 10^4 t^{2.107} \tag{1}$$

$$\frac{dP(t)}{dt} = 3.150 \times 10^3 + 101.7t^{2.107} - 3.232 \times 10^{-7}P(t) \tag{2}$$

$$\frac{dS(t)}{dt} = 3.232 \times 10^{-7}P(t) \tag{3}$$

where $w(t)$ (MT) is the annual plastic production at the eclipsed year $t$ since 1950, $P(t)$ (MT) is the cumulative amount of PS accumulated in the ocean at eclipsed year since 1950, $t$, $S(t)$ (MT) is the total amount of SOs in the ocean at the year $t$.

Equation (1) is a fitting result to the statistical data of global plastic production [1,2]. In Equation (3), $3.240 \times 10^{-7}$ (year$^{-1}$) is the rate of ST formation at 30 °C that equals to the rate of SOs formation.

Equation (2) was given by multiplying Equation (1) by the share of PS 7% and inflow ratio 3%. Equation (3) is based on the kinetic parameters of ST formation at 30 °C.

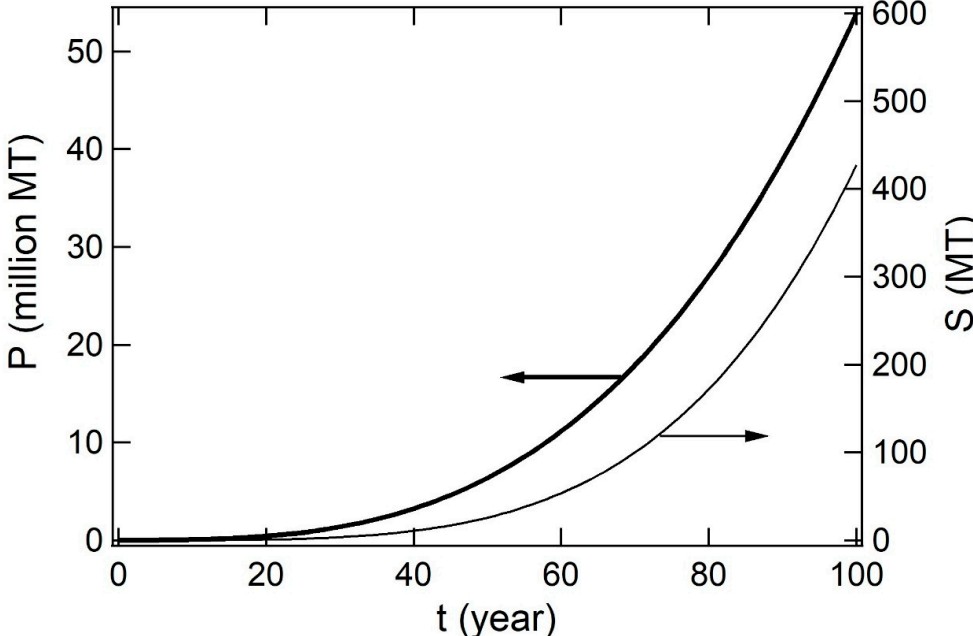

**Figure 6.** Simulated amounts of (P, left) PS debris and generated (S, right) SOs that have been accumulated in oceans at time *t* (year) since 1950 to 2050.

ST gradually decomposed to generate SM and SD and SD generated SM [22]. PS had been considered non-degradable in the nature, but kinetically PS, which contains no additives such as stabilizers, has been shown to degrade under the environmental temperature range to produce SOs. SOs formation would be enhanced under the severer conditions in the nature such as UV irradiation and exposure to oxygen in the air.

There has been the common misconception that plastic is stable and does not decompose at ambient temperatures. Accordingly, plastics have been considered to remain permanently intact in oceans. However, this study has clearly indicated this not to be the case, based on careful examination of PS degradation. In this study, even purified PS decomposition was shown to start even at 30 °C. Not only dose drifting lumps of PS break into micro/nano pieces, but also subsequently degrades into basic structure units.

It is widely known that the reaction rate doubles with a temperature increase of 10 °C. Tropical and subtropical ocean temperatures reach 30 °C and coastal sand at these latitudes reach 60 °C. The amounts of SO in the world oceans should thus be considered to be significantly higher than this PS value, suggesting ocean contamination to intensify as a result of plastics. Increasing amounts of various polymers go into the ocean and coastal area. Intensive studies are required to evaluate the biologic impacts of SOs from PS and the other chemical species possibly derived from various plastics.

## 4. Conclusions

SOs were often detected in sea water and sand of coastal area. PS decomposition at a low temperature range, 30 to 150 °C, was conducted to confirm SOs formation from PS wastes in the nature. In this temperature range, ST was a dominant product. ST would be an intermediate yielding $SD_1$, $SD_2$ and SM because SM became dominant in PS decomposition at the higher temperature range. The Arrhenius plot of ln *k* with the reciprocal of a reaction temperature in Kelvin showed a linear relationship within the low temperature range. The rate coefficient, *k* (year$^{-1}$), was obtained as 5.177

$exp(-5029/T)$ based on the rate of ST formation. SOs in the ocean was estimated over 400 MT in 2050 based on the simulation including thermal decomposition at 30 °C.

**Author Contributions:** Conceptualization, H.K., Y.K. and K.S.; data curation, H.K.; M.O. and K.S.; investigation, H.K., Y.K., K.K. and K.S.; writing of the original draft preparation, H.K., Y.K. and K.S.; writing of review and editing, H.K., Y.K., T.H. and K.Y. All authors have read and agreed to the published version of the manuscript.

**Funding:** This research received no external funding.

**Conflicts of Interest:** The authors declare no conflicts of interest.

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
