# Peer review of "Low Temperature Decomposition of Polystyrene"

_applsci, doi:10.3390/app10155100_

Round 1

Reviewer 1 Report

The manuscript describes a study of decomposition of purified polystyrene in the temperature range of 30-150ºC. The kinetics of the decomposition of polystyrene in a nitrogen atmosphere was studied and the decomposition products were identified by GC/MS. The authors confirmed that polystyrene can decompose into smaller molecules and release styrene oligomers and monomeric styrene in the process.

The authors suggest that their research was carried out to demonstrate that polystyrene does decompose in the ocean. However, the method that the authors used to degrade polystyrene does not quite reflect the reality of degradation of plastics in the ocean, as the decomposition experiment is carried out in bulk in the presence of PEG as a heat conducting medium with no effects of salty water, UV, etc. taken into account. Therefore, the reality of degradation of plastics in the ocean might be very different from what the results of the model experiment show. It would be good if the authors could address this issue in their manuscript.

In addition, the following issues should be addressed:

Section 3.4 titled Decomposition mechanism is basically a review, a summary of the previous works on the decomposition of polystyrene. Therefore, I was wondering why it is located in the Results and Discussion section where the original research is normally discussed.

The text in Figures 1, 2, and 3 is low resolution. Additionally, the spectra in Figure 3 are low resolution, therefore, the figure seems to be blurry and hard to read. It would be good if the quality of figures was improved.

The information given in Table 2 doubles that provided in Figure 4 (Arrhenius plot), therefore there is no reason to include both the figure and the table in the manuscript.

There are a few mistypes in the manuscript, such as KIA-RCT instead of IKA RCT in line 78, etc. The manuscript needs some additional proof reading to address this issue.

The English language of the manuscript is not quite up to the publishing standard. It would be good if the quality of English was improved throughout the whole text.

Therefore, I would recommend this manuscript for publication with a major revision.

Author Response

We appreciate the reviewers for sharing the precious time to review our paper.  And we do apologize the careless errors in the previous manuscript with figures of low resolutions bothering reviewers.

Please find the response in the attachment.

Reviewer 2 Report

The paper is clear, appropriate in length and well written. Some technical and text layout work is needed before final submission. Diagrams are not readable and should be redmade in Word (.docx). The causal-mechanical model was clear and appropriate. The topic is of interest. What we miss for this study  is to introduce variation in the independent variables as a cause of variation in the dependent. There is some variation in the effects of temperature and time but we cannot see clearly its influence in the variation of the dependent. We just find a sentence: "The effects of temperature and reaction time on PS decomposition at each temperature are shown in another paper". This is not acceptable. Please do provide for adequate data and for adequate statistical analysis thereof. This journal is not a second tier journal where you drop the rest of the results from a paper published somewhere else.

Author Response

First, we appreciate the reviewers for sharing the precious time to review our paper.  And we do apologize the careless errors in the previous manuscript with figures of low resolutions bothering reviewers.

Please find the response in the attachment.
